# Fast Image Super-Resolution Using Particle Swarm Optimization-Based Convolutional Neural Networks

**DOI:** 10.3390/s23041923

**Published:** 2023-02-08

**Authors:** Chaowei Zhou, Aimin Xiong

**Affiliations:** 1School of Physics and Telecommunication Engineering, South China Normal University, Guangzhou 510006, China; 2Qingyuan HuaYun Smart Control Technology Co., Ltd., Qingyuan 513200, China; 3SCNU Qingyuan Institute of Science and Technology Innovation Co., Ltd., Qingyuan 511517, China

**Keywords:** convolution neural network, particle swarm optimization, pneumonia diagnosis, super-resolution

## Abstract

Image super-resolution based on convolutional neural networks (CNN) is a hot topic in image processing. However, image super-resolution faces significant challenges in practical applications. Improving its performance on lightweight architectures is important for real-time super-resolution. In this paper, a joint algorithm consisting of modified particle swarm optimization (SMCPSO) and fast super-resolution convolutional neural networks (FSRCNN) is proposed. In addition, a mutation mechanism for particle swarm optimization (PSO) was obtained. Specifically, the SMCPSO algorithm was introduced to optimize the weights and bias of the CNNs, and the aggregation degree of the particles was adjusted adaptively by a mutation mechanism to ensure the global searching ability of the particles and the diversity of the population. The results showed that SMCPSO-FSRCNN achieved the most significant improvement, being about 4.84% better than the FSRCNN model, using the BSD100 data set at a scale factor of 2. In addition, a chest X-ray super-resolution images classification test experiment was conducted, and the experimental results demonstrated that the reconstruction ability of this model could improve the classification accuracy by 13.46%; in particular, the precision and recall rate of COVID-19 were improved by 45.3% and 6.92%, respectively.

## 1. Introduction

Single-image super-resolution (SR) refers to the reconstruction of low-resolution (LR) images to recreate high-resolution (HR) images as realistic as possible [1]. It has a promising future in medical and remote sensing, visual surveillance, etc. [2,3,4]. SR approaches used to be based on interpolation [5] and degradation [6] models. Currently, learning-based method [1,7] have received wide attention, among which deep learning models have shown powerful performance in image SR [8].

Convolutional neural networks (CNN) are widely used in image SR models. Many CNN-based methods attempt to learn how to achieve a better reconstruction performance by using deeper networks [9,10]. They have shown powerful performance in image SR; however, the high computing costs make them inconvenient for handling real-time problems. Dong et al. [11] proposed a lightweight model called FSRCNN, which has quite comparable performance to and is up to 40 times faster than SRCNN-EX [1]. Studies have further explored methods to improve the quality of SR images generated by FSRCNN. Considering the FSRCNN [11] model, it uses stochastic gradient descent (SGD) [12] optimization-based CNNs. However, the optimization problem for SR is non-convex and thus sensitive to the initial location. Such feature may cause the traditional neural network models to easily fall into the local optimum [13,14]. Particle swarm optimization (PSO) [15] has become a commonly used optimization method for training neural networks because of its simple rules and high search speed [16,17]. Kennedy et al. [18] used PSO to optimize the weight of a feedforward neural network, proposing the first combination of PSO and a neural network. Dong et al. [14] presented a modified PSO combined with an information entropy function to optimize the weights and bias of a back propagation neural network. The results showed that the joint algorithm had better performance in terms of accuracy and stability. Tu et al. [13] proposed an evolutionary convolutional neural network, which uses ModPSO and the backpropagation algorithm, to train convolutional neural networks to avoid models falling into local minima. This is the most advanced attempt; however, this study did not optimize particle swarm in terms of population diversity.

In order to further explore the effect of the PSO algorithm on CNN training and SR image quality, in this paper, a CNN training method based on the PSO algorithm was constructed, that is, the PSO algorithm was used to optimize CNN network parameters. In addition, in view of issues associated with PSO, the mutation of particles with high similarity is proposed according to the cosine similarity between particles, and the mutation probability decreased linearly with the number of iterations. The cosine similarity mutation strategy reinitialized the aggregated particles according to the cosine similarity, which could maintain a better spatial solution distribution of the particle swarm. Finally, the model was used to perform SR on low-resolution chest X-ray (CXR) images and analyze its impact on the diagnosis of pneumonia. The CXR images classification experiment showed that although the hybrid model was trained on a 91-image dataset, it could also super-resolve CXR images effectively and enhance the accuracy of their classification.

## 2. Related Work

### 2.1. Deep Learning for SR

Dong et al. [1] first proposed the use of a CNN model for image SR, which is called SRCNN. It is a lightweight model with three layers of construction but demonstrates advance repair qualities. It preprocesses the images using bicubic interpolation and then reconstructs SR images through nonlinear mapping of a three-layer convolutional neural network.

At that time, SRCNN was superior to all other reconstruction methods, but its interpolation structure would lead to too much computation when processing large images [11,19]. To speed up the process, Dong et al. [11] proposed the fast super-resolution convolutional neural networks (FSRCNN) model, which is 40 times faster than the SRCNN-EX model. In the FSRCNN model, the mean square error (MSE) is used as the cost function, which is formulated as:$$J(\theta )=\frac{1}{n}\sum_{i=1}^{n}||F(Y^{i};\theta )-X^{i}|{|}_{2}^{2}$$
where $X^{i}$ and $Y^{i}$ are the HR and LR training data pair, and $F(Y^{i};\theta )$ denotes the neural network output of the parameters θ. The goal of the stochastic gradient descent algorithm is to enforce the approach to 0. Its iteration formula is:$$\frac{\partial }{\partial \theta_{j}}J(\theta )=\frac{2}{n}(F(Y;\theta )-X)Y_{j}$$
$$\theta^{t+1}=\theta^{t}-\alpha \cdot \frac{\partial J(\theta )}{\partial \theta }$$
where α represents the step size of each update, usually 0.01, 0.001, and 0.0001

### 2.2. PSO Based on Centroid Opposition-Based Learning

PSO is an interaction-based optimization algorithm imitating the preying behavior of birds [15]. Each particle delegates a group of weights and bias, and the optimal solution is obtained by iteratively searching particles in the solution space. The particles update their position in two ways: one is the individual optimal solution (pbest), and the other is the global group optimal solution (gbest). The dynamical formulas of PSO are as follows:(1)$$v_{i,d}^{k+1}=wv_{i,d}^{k}+c_{1}r_{1}(pbest_{i,d}^{k}-x_{i,d}^{k})+c_{2}r_{2}(gbest_{d}^{k}-x_{i,d}^{k})$$
(2)$$x_{i,d}^{k+1}=x_{i,d}^{k}+v_{i,d}^{k+1}(i=1,2,\ldots ,N)$$
where N is the initial number of particles, $v_{i}^{k}$ and $x_{i}^{k}$ measure the velocity and position of the $i^{th}$ particle at the $k^{th}$ iteration, respectively, ω denotes the inertia weight, which reflects the effect of the previous velocity on the current velocity, $c_{1}$ and $c_{2}$ are acceleration factors, usually represented by two real numbers, $r_{1}$ and $r_{2}$ are unpremeditated numbers from the interval (0,1). 

Opposition-based learning (OBL) [20] was proved to be an effective means to improve particle swarm optimization algorithms [21]. The central idea of OBL is to improve the optimization ability by searching a solution and its corresponding opposing solution simultaneously in the solution space. Centroid opposition-based learning (COBL) [22] makes use of the centroid of the swarm when calculating the position of opposing solutions and utilizes the swarm experience to improve the searching efficiency of the particle swarm. Assuming that ($X_{1},X_{2},\ldots ,X_{n}$) is the location of *n* particles, the centroid is calculated as follows:$$M=\frac{1}{n}\sum_{i=1}^{n}x_{i}$$

The opposite solution based on the centroid of the swarm can be formulated as: $$ox_{i}=2M-x_{i}$$

The opposite solution exists in a search space with dynamic boundaries, which is expressed as:$$a_{j}=\min(x_{i,j})\;\;\;b_{j}=\max(x_{i,j})$$

If the opposite solution exceeds the dynamic boundary, the opposite solution is recalculated according to the following equation:$$x_{i}=\left\{\begin{array}{l} \min(x_{i})+rand\;\;(M-\min(x_{i})),ox_{i}<a\\ M+rand\;\;(\max(x_{i})-M),ox_{i}>b \end{array}\right.$$

By comparing the current solution with the opposite solution, the better one can be selected. 

## 3. Methods

### 3.1. Cosine Similarity Variation Strategy

In the late phase of COBL, the algorithm will be trapped into a local optimum due to the extreme population aggregation. Cosine similarity is mainly used to measure the size of the difference between two individuals, and here it is used to quantitatively describe the aggregation degree of particles and populations, according to the formulas:$$\mathrm{Cos}(gbest,x_{i})=\frac{gbest\cdot x_{i}}{||gbest|{|}_{2}||x_{i}|{|}_{2}}$$
$$\mathrm{Cos}\_a=\frac{1}{n}\sum_{i=1}^{n}\mathrm{Cos}(gbest,x_{i})$$
where gbest represents the global optimal solution of the current iteration, and $x_{i}$ denotes the location of the $i^{th}$ particle. In order to improve the search efficiency, some particles, whose cosine similarity is greater than the average cosine similarity of the population, are re-initialized to maintain the diversity of the particles. The average cosine similarity was calculated by the cosine similarity between each particle and pbest, and the mutation region was defined according to the average cosine similarity, as indicated by the red dotted line in Figure 1. The region whose cosine similarity to pbest is greater than the average similarity is the mutation region, and the particles in this region have a certain probability to be randomly initialized. The region whose cosine similarity to pbest is less than the average similarity is defined as the non-mutation region, and the particles in this region continue the iterative optimization.

In addition, the cosine similarity of the population is affected by some extreme particles, which may lead to a higher mutation rate. However, a higher mutation rate may be detrimental to the later convergence of the algorithm. To address this issue, A mutation factor δ is introduced, which reduces the probability of mutation from 50% to 5% as the number of iterations increases. The mutation factor can be defined as:$$\delta =M_{start}-\frac{M_{start}-M_{end}}{T_{\max}}\cdot k$$
where $M_{start}$ and $M_{end}$ represent the initial and final mutation probability, respectively, k indicates the number of current iterations, and $T_{\max}$ denotes the limit number of iterations.

### 3.2. FSRCNN Model Based on SMCPSO

In our implementation, we utilized the SMCPSO method to initialize the weights and bias of the FSRCNN model. The MSE is defined as the fitness function of SMCPSO, and the dimension of the particles is the number of parameters to be learned in the FSRCNN network. Figure 2 illustrates the flowchart of the joint algorithm. The weight and bias of the FSRCNN model correspond to each dimension of the particle. The number of optimized particles was set to 50, and each particle represented a set of possible weights and biases of the FSRCNN model. The number of iterations was set to 10,000. Every 100 iterations, the particle whose cosine similarity to the optimal particle was less than the average cosine similarity and whose random value was greater than the variation factor was initialized. Each iteration considers whether there is a better solution for the inverse particle of the particle, and if so, transforms the particle into its inverse particle. When the iteration stop condition is reached, the particle swarm training ends. The value of each dimension of the optimal particle corresponds to the weight and bias of the FSRCNN model; the SGD algorithm was used to optimize the weight and bias of the model until the training was completed. The SGD algorithm is greatly affected by the initial position; therefore, PSO can set the ideal initial position for SGD. Specifically, PSO is used to search the desired weights and bias of the CNN as the initial parameters of the SGD algorithm. Higher accuracy can be achieved through this joint training method.

### 3.3. Classification of Pneumonia

Deep convolutional neural networks are being used in medical diagnosis. ResNet34 [23] adopted the residual network structure to achieve a good balance between classification accuracy and network complexity. Therefore, ResNet34 [23] was selected as the diagnostic classifier for pneumonia. Since the pneumonia data set publicly available online is not large, transfer learning was used to train the model. ResNet34 [23] uses ImageNet weights, and the full connection layer was modified to fit the four categories of the experimental data set.

Five indexes, i.e., accuracy, precision, sensitivity, F1 score, and specificity [24], were used to evaluate the classification results of ResNet34 [23]. The calculation formulas are as follows:$$accuracy_{class_{i}}=\frac{TP_{class_{i}}+TN_{class_{i}}}{TP_{class_{i}}+TN_{class_{i}}+FP_{class_{i}}+FN_{class_{i}}}$$
$$precision=\frac{TP_{class_{i}}}{TP_{class_{i}}+FP_{class_{i}}}$$
$$sensitivity_{class_{i}}=\frac{TP_{class_{i}}}{TP_{class_{i}}+FN_{class_{i}}}$$
$$F1\_score_{class_{i}}=2\frac{precision_{class_{i}}\times sensitivity_{class_{i}}}{precision_{class_{i}}+sensitivity_{class_{i}}}$$
$$specificity=\frac{TN_{class_{i}}}{TN_{class_{i}}+FP_{class_{i}}}$$

We considered COVID-19, Lung Opacity, Normal, and Viral Pneumonia as the four class problems.

## 4. Experiments and Results

### 4.1. Improved Particle Swarm Optimization

For the sake of evaluating the search capability of the proposed SMCPSO algorithm, 15 multimodal test functions (F6–F20), recommended by IEEE Evolutionary Computing Conference (CEC) 2013 [25], were used to test the algorithm. The optimal solutions of the F6–F20 functions are −900, −800, −700, −600, −500, −400, −300, −200, −100, 100, 200, 300, 400, 500, 600, 700, respectively. In the experiment, the population size was N = 40, the maximum number of function evaluations was 10,000, and the problem dimension was D = 30. The standard PSO, COBL, and SMCPSO algorithms were used in 20 independent tests. Table 1 records the best value, worst value, mean value, standard deviation, mean absolute error, and time of each algorithm. The results in the Table 1 show that SMCPSO could find better solutions for F6–F20 multi-peak test functions. Especially for the F6, F10, F14, and F19 functions, the optimization of PSO and COBL was not satisfactory, but SMCPSO found a solution extraordinarily close to the global optimal solution.

As shown in Figure 3, the SMCPO demonstrated improving searching ability for most test functions, and only a slightly lower ability compared to COBL in the initial stage. This is because in the early stage of the SMCPSO algorithm search, the particles with high cosine similarity to the pbest mutated in the optimization process, which was not conducive to the algorithm convergence in the early stage; however, this could make the algorithm less likely to be confused by the local optimal solution. With the increase of the function evaluation times, the SMCPSO algorithm showed a better global searching ability than the standard PSO and COBL models in most multimodal optimization problems.

### 4.2. FSRCNN Model Based on SMCPSO

Consistent with the work of SRCNN and FSRCNN, a 91-image dataset [26] was used for training, and Set5 [27], Set14 [28], BSD100 data set [29] and Urban data set [30] were used for testing. The peak signal-to-noise ratio (PNSR) [31] and the structural similarity index (SSIM) [32] were employed to evaluate the quality of the images.
$$\mathrm{PNSR}=10\log_{10}(\frac{{\left(2^{n}-1\right)}^{2}}{\mathrm{MSE}})$$
where
$$\mathrm{MSE}=\frac{1}{H\times W}\sum_{i=1}^{H}\sum_{j=1}^{W}{\left(X(i,j)-Y(i,j)\right)}^{2}$$
$$\mathrm{SSIM}=\frac{\left(2u_{X}u_{Y}+C_{1})(2\sigma_{XY}+C_{2}\right)}{\left(u_{X}^{2}+u_{X}^{2}+C_{1})(\sigma_{X}^{2}+\sigma_{Y}^{2}+C_{2}\right)}$$
where $u_{X}$ and $u_{Y}$ represent the mean values of images *X* and *Y*, respectively, and were used for the estimation of image brightness; $\sigma_{X}$ and $\sigma_{Y}$ denote the standard deviations of images *X* and *Y*, respectively, and were used for the estimation of contrast; $\sigma_{XY}$ represents the covariance of images *X* and *Y*, and was used for the measurement of structural similarity. *C*_1_ and *C*_2_ are constants that prevent the denominator from being 0.

SMCPSO was used to optimize FSRCNN and compared with SGD optimization; their performances were compared using Set5 [27]. In Figure 4a,b, compared with the SGD method, the images generated by the SMCPSO-SGD method show better performance as regards the PNSR [31] and the SSIM [32]. Moreover, the improvement of PNSR and SSIM with SMCPSO-SGD training was relatively stable.

In Figure 4c, the training loss function of the SMCPSO-SGD method converges fast and shows a small error; in Figure 4d, the eval loss function of the SMCPSO-SGD method declines stably and shows a lower error. These results showed that SMCPSO can help the SGD algorithm find better solutions faster.

To verify the universality of the algorithm, image quality was evaluated using four general test sets. Table 2 demonstrates the test results of the SMCPSO-SGD using the general test sets Set5 [27], Set14 [28], BSD100 data set [29], and Urban data set [30]. The results indicated that the model trained with SMCPSO performed best as regards PNSR and SSIM at all scales. With the BSD100 dataset, when the scale factor was 2, the improvement of the SMCPSO-SGD model was the most significant, about 1.54 db, compared with SGD model. This suggests that the SMCPSO optimization algorithm can help models generate more realistic images.

### 4.3. Classification Evaluation

To confirm the effectiveness of generating image details, we conducted an experiment on chest X-ray (CXR) SR image classification. The data set we used was from Kaggle (available on the website: https://www.kaggle.com/datasets/tawsifurrahman/covid19-radiography-database (accessed on 19 March 2022)) [24]. It has a total of 21,165 CXR images, divided in four categories. The “COVID-19” and “Viral Pneumonia” classes were data-enhanced to balance the training set. Table 3 summarizes the number of images per class used for training, validation, and testing. CXR LR images for each category were obtained from CXR HR images using the down-sampling method. Then, the CXR LR images, CXR SR images, and CXR HR images were fed into the ResNet34 [23] classification network. The implementation scheme is shown in Figure 5. 

In order to evaluate the reliability of the restored details, the ResNet34 [23] classification network was used to classify LR CXR images, SR CXR images, and HR CXR images. Five performance indicators, i.e., accuracy, precision, sensitivity, F1 score, and specificity [24], were used to evaluate the classification results. Table 4 shows the evaluation results.

Figure 6 displays the CXR HR, LR, and SR images; the CXR SR image was recovered from the CXR LR image using the SMCPSO-SGD model. In terms of visual perception, images processed by SMCPSO-SGD had better visual perception and higher scores of PNSR and SSIM. Although the given CXR LR images lost a large amount of valid information, this model could help acquire a better visual effect. Figure 7 shows the matrix diagram of CXR HR, LR, and SR image classification for subsequent diagnosis using the ResNet34 model. It was found that the CXR SR images improved the diagnostic accuracy of COVID-19, lung opacity, normal and viral pneumonia from 80.89% to 90.03%, 39.10% to 80.53%, 65.75% to 90.58%, and 0.00% to 11.09%, respectively. 

In order to further evaluate the comprehensive performance of the algorithm, the five indexes of accuracy, precision, sensitivity, F1 scores, and specificity were used to evaluate the classification results. As can be seen in Table 4, the CXR LR image classification accuracy was the lowest, indicating that the LR images contained very little useful information conducive to classification, while the SMCPSO-SGD model led to improvement in five evaluation indicators, increasing the CXR LR image classification accuracy to values close to those obtained with CXR HR images. The results revealed that the model could recover the images’ real and useful details well, which is beneficial to the diagnosis of pneumonia. 

## 5. Conclusions

A training method of a convolution neural network based on an improved particle swarm optimization algorithm was proposed. It was proved that the joint training method could improve the efficiency and accuracy of the model. A mutation strategy was proposed and proved to be able to prevent a premature improvement of the optimization ability of the particle swarm optimization algorithm. Moreover, the chest X-ray image classification experiment proved that the proposed model can reconstruct useful information for pneumonia recognition.

## Figures and Tables

**Figure 1 sensors-23-01923-f001:**
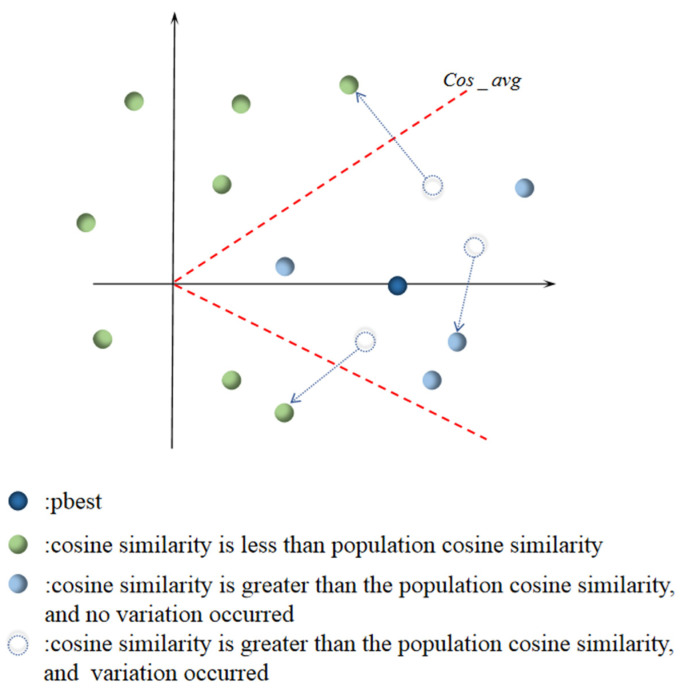
Schematic diagram of the cosine similarity mutation strategy. The red dotted line represents the average cosine similarity of the population, and the dotted arrow shows one possible position of the particle after the variation.

**Figure 2 sensors-23-01923-f002:**
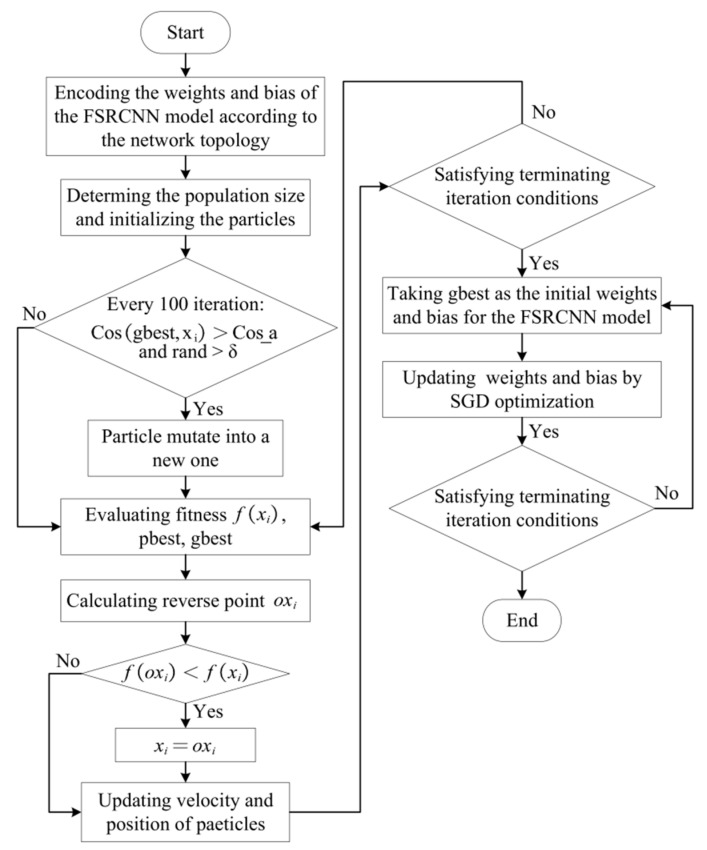
Flowchart of SMCPSO-based CNNs.

**Figure 3 sensors-23-01923-f003:**
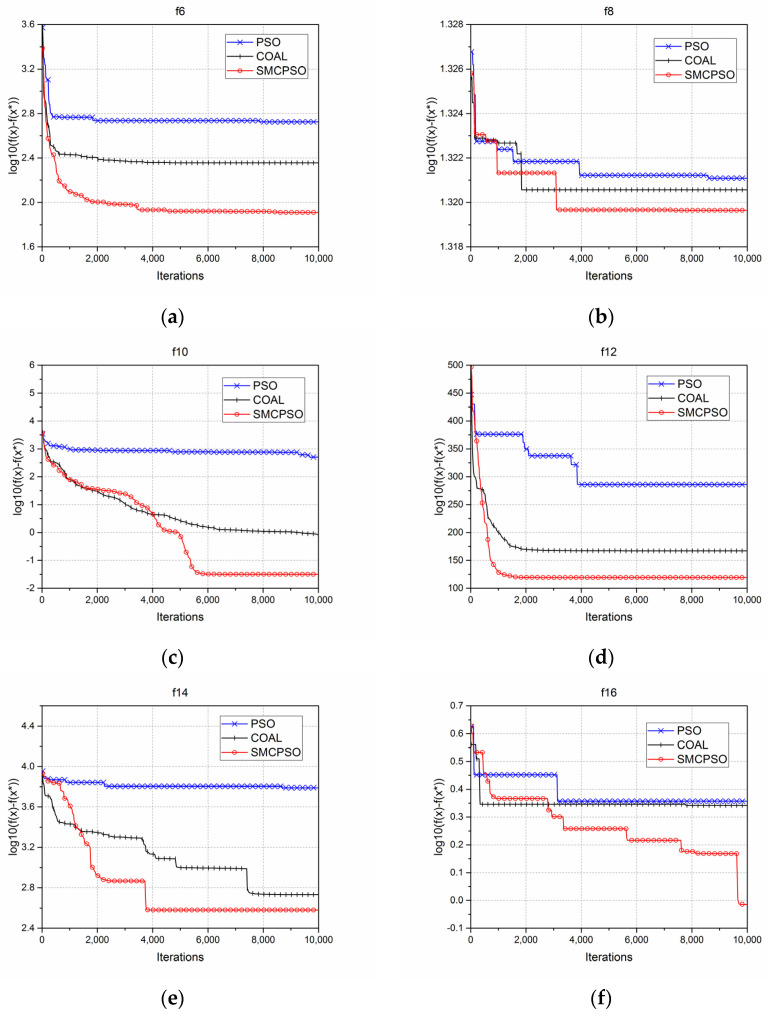
Comparison of the results for PSO, COBL, and SMCPSO in relation to multi-peak test functions. (f(x) represents the optimal solution of the current iteration, and f(x∗) represents the actual optimal solution of the function. f(x)−f(x∗) is the current error represented by a logarithm, and the smaller the error is, the better the algorithm performance is.) (**a**) Errors between the optimal value of each iteration and the actual optimal value of the f6 multi-peak test function by the standard PSO, COBL, and SMCPSO methods, respectively; similarly, this is shown for (**b**) f8, (**c**) f10, (**d**) f12, (**e**) f14, (**f**) f16, (**g**) f18, (**h**) f20.

**Figure 4 sensors-23-01923-f004:**
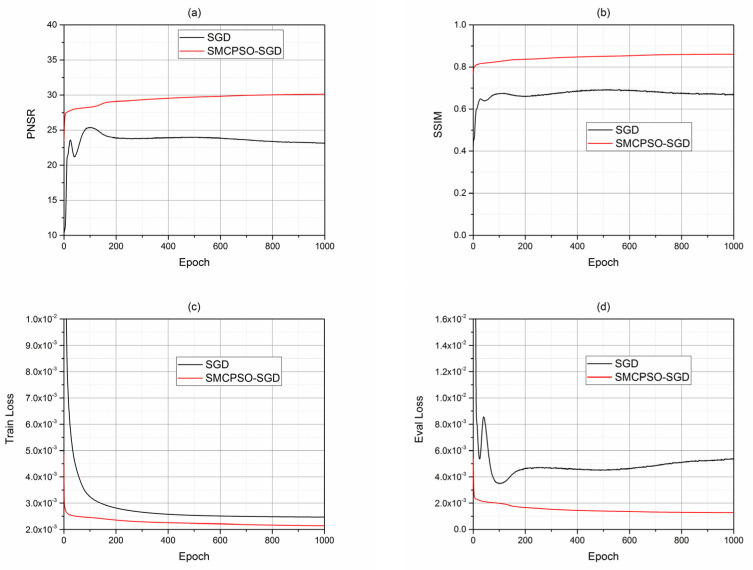
Comparison of SGD-alone training and SMCPSO-SGD combined training with SET5. In (**a**,**b**), PNSR and SSIM of images generated during SGD and SMCPSO-SGD training are compared. The red curve is smoother than the black curve, and the PNSR and SSIM of the images are higher in each epoch, indicating better quality of the generated image. In (**c**,**d**), the train loss and eval loss of the red curve are always lower, and the black curve declines steadily in (**c**) but fluctuates greatly in (**d**), while the red curve declines steadily all the time, indicating that SMCPSO helps SGD to better train.

**Figure 5 sensors-23-01923-f005:**
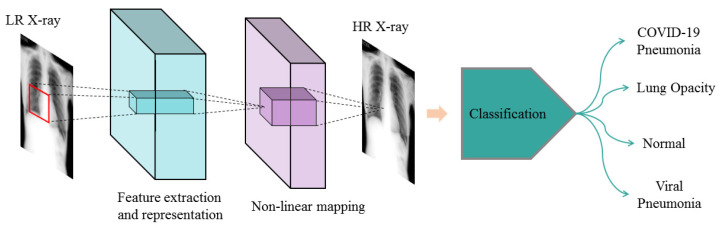
Pneumonia diagnosis flow diagram. (The SMCPSO-SGD network was first used to convert LR X-rays into HR X-rays, then ResNet34 was used to classify the HR X-ray images).

**Figure 6 sensors-23-01923-f006:**
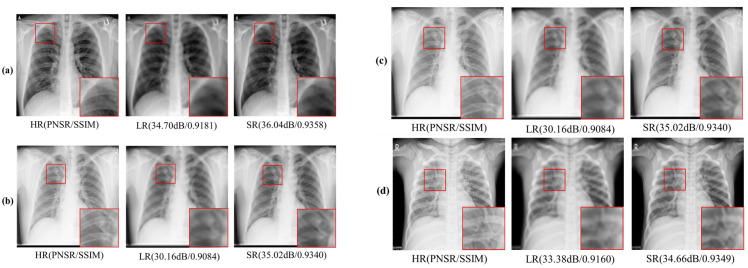
Examples of CXR HR, LR, and SR images for the COVID-19 category (**a**), Lung Opacity category (**b**), Normal category (**c**), Viral Pneumonia category (**d**).

**Figure 7 sensors-23-01923-f007:**
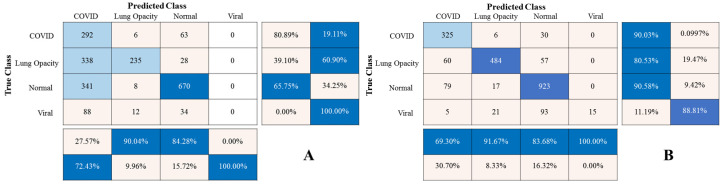
Classification mixing matrix for three kinds of images. (The vertical axis represents the true category, the horizontal axis represents the predicted category, the ratio matrix on the right represents the accuracy of each of the four categories, and the lower ratio matrix represents the recall rate of each of the four categories.) (**A**) Classification result for the CXR LR images. Since the CXR LR images were obtained from CXR HR images by using the down-sampling method, there was a loss of information, resulting in the worst classification. (**B**) Classification results for the CXR SR images. The SMCPOS-SGD model was used to process the CXR LR images, and the details of the CXR LR images could be restored. The generated CXR SR images could effectively improve the number of correct classifications for the four categories. (**C**) Classification results for the CXR HR images. The CXR HR images are the most primitive images; therefore, it is important to classify them correctly.

**Table 1 sensors-23-01923-t001:** Statistical summary of the best value, worst value, average value, standard deviation, mean absolute error, and search time for the PSO, COBL, and SMCPSO algorithms in relation to 15 multimodal test functions (F6–F20) recommended by the IEEE Evolutionary Computing Conference (CEC) 2013 [25] (the optimal solutions of the F6-F20 functions are −900, −800, −700, −600, −500, −400, −300, −200, −100, 100, 200, 300, 400, 500, 600, 700, respectively).

Fun	Alg	Best	Worst	Mean	Std	Mae	Time
	PSO	−6.86 × 10^2^	−3.61 × 10^2^	−5.53 × 10^2^	7.89× 10	3.47 × 10^2^	5.18
F6	COBL	−754 × 10^2^	−4.83 × 10^2^	−5.89 × 10^2^	7.10 × 10	3.11 × 10^2^	6.31
	SECPSO	−9.00 × 10^2^	−7.55 × 10^2^	−8.20 × 10^2^	2.35 × 10	8.05 × 10	7.24
	PSO	−7.12 × 10^2^	−6.47 × 10^2^	−6.79 × 10^2^	1.87 × 10	1.21 × 10^2^	9.12
F7	COBL	−7.57 × 10^2^	−6.53 × 10^2^	−7.09 × 10^2^	2.53 × 10	9.07 × 10	15.34
	SECPSO	−7.58 × 10^2^	−6.95 × 10^2^	−7.31 × 10^2^	1.81 × 10	6.90 × 10	17.23
	PSO	−6.79 × 10^2^	−6.79 × 10^2^	−6.79 × 10^2^	4.67 × 10^−2^	2.09 × 10	7.34
F8	COBL	−6.79 × 10^2^	−6.79 × 10^2^	−6.79 × 10^2^	6.34 × 10^−2^	2.09 × 10	10.85
	SECPSO	−6.79 × 10^2^	−6.79 × 10^2^	−6.79 × 10^2^	5.23 × 10^−2^	2.08 × 10	13.33
	PSO	−5.71 × 10^2^	−5.61 × 10^2^	−5.66 × 10^2^	2.36	3.41 × 10	61.09
F9	COBL	−5.77 × 10^2^	−5.70 × 10^2^	−5.73 × 10^2^	2.52	2.71 × 10	90.7
	SECPSO	−5.77 × 10^2^	−5.66 × 10^2^	−5.72 × 10^2^	3.58	2.83 × 10	108.47
	PSO	−2.53 × 10^2^	3.74 × 10^2^	1.06	1.54 × 10^2^	5.01 × 10^2^	6.45
F10	COBL	−4.25 × 10^2^	2.46 × 10^2^	−2.35 × 10^2^	1.81 × 10^2^	2.65 × 10^2^	7.99
	SECPSO	−5.00 × 10^2^	−5.00× 10^2^	−5.00 × 10^2^	1.10 × 10^−1^	1.42 × 10^−1^	10.62
	PSO	−1.51 × 10^2^	−7.53 × 10	−1.19 × 10^2^	2.05 × 10	2.81 × 10^2^	6.97
F11	COBL	−2.74 × 10^2^	−1.78 × 10^2^	−2.35 × 10^2^	2.51 × 10	1.65 × 10^2^	8.43
	SECPSO	−3.90 × 10^2^	−3.66 × 10^2^	−3.77 × 10^2^	7.21	2.30 × 10	11.77
	PSO	−7.63 × 10	1.26 × 10	−2.28 × 10	2.29 × 10	2.77 × 10^2^	7.99
F12	COBL	−2.03 × 10^2^	−2.74 × 10	−1.43 × 10^2^	4.10 × 10	1.57 × 10^2^	10.66
	SECPSO	−2.26 × 10^2^	−8.61 × 10	−1.59 × 10^2^	4.07 × 10	1.41 × 10^2^	15.08
	PSO	2.10 × 10	1.24× 10^2^	8.03 × 10	2.21 × 10	2.80 × 10^2^	7.75
F13	COBL	−2.58 × 10	5.11 × 10	7.34	2.69 × 10	2.07 × 10^2^	10.96
	SECPSO	−8.52 × 10	2.05 × 10	−1.66 × 10	3.70 × 10	1.83 × 10^2^	14.74
	PSO	5.86 × 10^3^	6.87 × 10^3^	6.46 × 10^3^	2.65 × 10^2^	6.56 × 10^3^	6.99
F14	COBL	3.39 × 10	4.73 × 10^2^	1.99 × 10^2^	1.18 × 10^2^	2.99 × 10^2^	9.41
	SECPSO	−1.08 × 10	4.77 × 10^2^	1.97 × 10^2^	1.93 × 10^2^	2.97 × 10^2^	15.07
	PSO	6.41 × 10^3^	7.69 × 10^3^	7.20 × 10^3^	2.87 × 10^2^	7.10 × 10^3^	7.98
F15	COBL	3.26 × 10^3^	7.11 × 10^3^	5.26 × 10^3^	1.25 × 10^3^	5.16 × 10^3^	10.51
	SECPSO	2.84 × 10^3^	4.99 × 10^3^	4.04 × 10^3^	7.36 × 10^2^	3.94 × 10^3^	15.25
	PSO	2.02 × 10^2^	2.03 × 10^2^	2.02 × 10^2^	2.86 × 10^−1^	2.28	37.05
F16	COBL	2.02 × 10^2^	2.02 × 10^2^	2.02 × 10^2^	2.08 × 10^−1^	2.04	69.27
	SECPSO	2.01 × 10^2^	2.02 × 10^2^	2.01 × 10^2^	3.93 × 10^−1^	1.47	80.66
	PSO	5.88 × 10^2^	7.54 × 10^2^	6.69 × 10^2^	4.07 × 10	3.69 × 10^2^	5.24
F17	COBL	4.08 × 10^2^	4.76 × 10^2^	4.38 × 10^2^	1.74 × 10	1.38 × 10^2^	7.17
	SECPSO	3.63 × 10^2^	4.06 × 10^2^	3.90 × 10^2^	1.29 × 10	9.25 × 10	10.96
	PSO	7.06 × 10^2^	8.34 × 10^2^	7.76 × 10^2^	3.22 × 10	3.76 × 10^2^	6.45
F18	COBL	5.00 × 10^2^	6.65 × 10^2^	5.91 × 10^2^	4.83 × 10	1.91 × 10^2^	8.85
	SECPSO	4.99 × 10^2^	6.31 × 10^2^	5.54 × 10^2^	4.17 × 10	1.53 × 10^2^	11.8
	PSO	5.50 × 10^2^	8.76 × 10^2^	6.04 × 10^2^	9.81 × 10	7.04 × 10^2^	5.49
F19	COBL	5.43 × 10^2^	6.67 × 10^2^	5.84 × 10^2^	2.92 × 10	8.36 × 10	9.31
	SECPSO	5.04 × 10^2^	5.11 × 10^2^	5.09 × 10^2^	2.12	6.65	9.32
	PSO	6.12 × 10^2^	6.14 × 10^2^	6.13 × 10^2^	3.95 × 10^−1^	1.33 × 10	6.54
F20	COBL	6.11 × 10^2^	6.13 × 10^2^	6.12 × 10^2^	6.28 × 10^−1^	1.22 × 10	9.18
	SECPSO	6.11 × 10^2^	6.12 × 10^2^	6.11 × 10^2^	6.17 × 10^−1^	1.17 × 10	8.74

**Table 2 sensors-23-01923-t002:** PNSR and SSIM of the bicubic, SGD, SMCPSO-SGD models with the test sets Set5 [27], Set14 [28], BSD200 [29], and Urban100 [30].

Model	Factor	Set5	Set14	BSD100	Urban
		PNSR/SSIM	PNSR/SSIM	PNSR/SSIM	PNSR/SSIM
Bicubic		33.66/0.9299	30.24/0.8688	29.56/0.8431	26.88/0.8403
SGD	×2	37.00/0.9558	32.63/0.9088	31.80/0.9074	-
SMCPSO		37.20/0.9590	32.67/0.9137	33.34/0.9214	29.91/0.9048
Bicubic		30.39/0.8682	27.55/0.7742	27.21/0.7385	24.46/0.7349
SGD	×3	33.16/0.9140	29.43/0.8242	28.60/0.8137	-
SMCPSO		33.77/0.9305	29.85/0.8479	29.03/0.8113	27.40/0.8340
Bicubic		33.66/0.9299	26.00/0.7027	25.96/0.6675	23.14/0.6577
SGD	×4	30.71/0.8657	27.59/0.7535	26.98/0.7398	-
SMCPSO		31.75/0.8755	27.69/0.7706	27.90/0.7535	24.63/0.7340

**Table 3 sensors-23-01923-t003:** Number of images per class.

Type	No. of Image	Train	Augmented	Valid	Test
COVID-19	3616	2894	2894	361	361
Lung Opacity	6012	4810	-	601	601
Normal	10,192	8154	-	1019	1019
Viral Pneumonia	1345	1077	4308	134	134

**Table 4 sensors-23-01923-t004:** Comparison of HR, LR, and SR image classification results.

	Accuracy	Precision	Sensitivity	F1 Scores	Specificity
CXR LR	76.79%	70.90%	56.60%	0.5810	0.8655
CXR SR	90.25%	84.87%	83.40%	0.8170	0.8992
CXR HR	96.03%	94.34%	94.18%	0.9417	0.9556

## Data Availability

Data is not available due to privacy restrictions.

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
