# Peer review of "Fast Image Super-Resolution Using Particle Swarm Optimization-Based Convolutional Neural Networks"

_sensors, 2023, doi:10.3390/s23041923_

Round 1

Reviewer 1 Report (Previous Reviewer 1)

The manuscript must be revised, after which it can be reconsidered by the journal for acceptance. Below are comments and suggestions to help the authors improve it:

1\ The whole reference list is NOT organized properly. After abbreviated journal title (italicized), there must be publication year (in bold font), then volume number (italicized), and finally page numbers.The authors probably did not pay any attention to how the others format references in MDPI journals.

2\ Caption to Fig.7: the authors must add description for each panel (A)-(C) here

3\ Fig.4, caption: description for each panel (a)-(d) must be added here

4\ Fig.3, caption: description for each panel (a)-(h) must be added here

Author Response

Response to Reviewer 1 Comments(The attachment is the same)

Point 1: The whole reference list is NOT organized properly. After abbreviated journal title (italicized), there must be publication year (in bold font), then volume number (italicized), and finally page numbers.The authors probably did not pay any attention to how the others format references in MDPI journals.

Response 1: Thank you for pointing out our mistake in detail. We immediately realized the problem and revised the reference part. Modified as follows:

References

  1. Dong,; Loy,C. C.; He, K.; Tang, X. Learning a Deep Convolutional Network for Image Super-Resolution. In Proceedings of European Conference on Computer Vision(ECCV), Zurich, Switzerland, September 6-12, 2014; pp.184-199.
  2. Huang,;Wang, L.; Qin, J.; Chen, Y.; Cheng, X.; Zhu, Y. Super-Resolution of Intravoxel Incoherent Motion Imaging Based on Multisimilarity. IEEE Sens. J. 202020, 10963-10973.
  3. Dong, ;Zhang, L.; Fu, H. RRSGAN: Reference-Based Super-Resolution for Remote Sensing Image. IEEE T. Geosci. Remote 2022, 60, 1-17.
  4. Ha, ;Tian, J.; Miao, Q.; Yang, Q.; Guo, J.; Jiang, R. Part-Based Enhanced Super Resolution Network for Low-Resolution Person Re-Identification. IEEE Access. 2020, 8, 57594-57605..
  5. Li, X.;Orchard, T. New Edge Directed Interpolation. In Proceedings of International Conference on Information Processing(ICIP), Vancouver, BC, Canada, 10-13 September 2000; pp. 311-314.
  6. Capel,; Zisserman, A.Super-Resolution Enhancement of Text Image Sequences. In Proceedings of 15th International Conference on Pattern Recognition(ICPR), Barcelona, Spain, 3-7 September 2000; pp. 311-314.
  7. Huang, ;Jiang, Z.; Lan, R.; Zhang, S.; Pi, K. Infrared Image Super-Resolution via Transfer Learning and PSRGAN. IEEE Signal Proc. Let. 2021, 28, 982-986.
  8. Shen, ; Hou,B.; Wen, Z.; Jiao, L. Structural-Correlated Self-Examples Based Super-Resolution of Single Remote Sensing Image. IEEE J. of Selected Topics in Applied Earth Observations and Remote Sensing. 2018, 11, 3209-3223.
  9. Hu, ; Li, T.; Zhao, M.; Ning, J. Hyperspectral Image Super-Resolution via Deep Structure and Texture Interfusion. IEEE J.of Selected Topics in Applied Earth Observations and Remote Sensing. 2021, 14, 8665-8678.
  10. Shang, ; Li, X.; Foody, G. M.; Du, Y.; Ling,F. Super-Resolution Land Cover Mapping Using a Generative Adversarial Network IEEE Geosci. Remote S. 202219, 1-5.
  11. Dong, C.; Loy, C. C.; Tang, X. Accelerating the Super-Resolution Convolutional Neural Network. In Proceedings of European Conference on Computer Vision(ECCV), Amsterdam, Netherlands, 8-16 October 2016; 391-407.
  12. Ruder, (National University of Ireland Galway, Galway, Ireland). An Overview of Gradient Descent Optimization Algorithms, 2017.
  13. Tu, S.; Waqas, M.; Shah, Z.; Rehman, O.; Yang, Z.; Koubaa,A.; Rehman, S. U. ModPSO-CNN: An Evolutionary Convolution Neural Network with Application to Visual Recognition. Soft Computer. 2021,25, 2165–
  14. Dong,X.; Lian, ; Liu, Y.Small and Multi-Peak Nonlinear Time Series Forecasting Using a Hybrid Back Propagation Neural Network. Inform. Sciences. 2017, 424, 39–54.
  15. Kennedy, ; Eberhart, R.Particle Swarm Optimization. In Proceedings of IEEE International Conference on Neural Networks(ICNN), Perth, WA, Australia, 27-30 November 1995; pp. 1942-1948.
  16. Peng,;Deng, W.; Wu, W.; Luo, Z.; Huang J. Hybrid Modeling Routine for Metal-Oxide TFTs Based on Particle Swarm Optimization and Artificial Neural Network. Electron. Lett. 202056, 453-456.
  17. Zhang,L.; Zhao, L. High-Quality Face Image Generation Using Particle Swarm Optimization-Based Generative Adversarial Networks. FutuGener. Comp. Syst. 2021, 122, 98-104.
  18. Kennedy, ; Eberhart, R.; Shi, Y.Swarm Intelligence. Publisher: Morgan Kaufmann Publishers, San Francisco, 2001.
  19. Kim, ; Lee, J. K.; Lee, K. M.Accurate Image Super-Resolution Using Very Deep Convolutional Networks. In Proceedings of IEEE Conference on Computer Vision and Pattern Recognition (CVPR), Las Vegas, NV, USA, 27-30 June 2016; pp. 1646-1654.
  20. Tizhoosh, R. Opposition-Based Learning: A New Scheme for Machine Intelligence. In Proceedings of International Conference on Computational Intelligence for Modelling, Control and Automation and International Conference on Intelligent Agents, Web Technologies and Internet Commerce(CIMCA), Vienna, Austria, 28-30 November 2005; pp. 695-701.
  21. Xu, ;Wang, L.; Wang, N.; Hei, X.;Zhao, L. A. Review of Opposition-Based Learning from 2005 to 2012. Engineering Applications of Artificial Intelligence. 2014, 29, 1-12.
  22. Rahnamayan, S.; Jesuthasan, J.; Bourennani, F.; Salehinejad, H.; Naterer, G. Computing opposition by involving entire population. In Proceedings ofIEEE Congress on Evolutionary Computation (CEC), Beijing, China, 6-11 July 2014; pp. 1800-1807.
  23. Liang, J. J.; Qu,B. Y.; Suganthan, P. N.; Hernández-Díaz, A. Problem Definitions and Evaluation Criteria for the CEC 2013 Special Session on Real-Parameter Optimization.Technical Report. 2013, 201212, 281–
  24. Yang, ; Wright,J.; Huang, T. S.; Ma, Y. Image Super-Resolution Via Sparse Representation. IEEE T. Image Process. 2010, 19, 2861-2873.
  25. Bevilacqua, ; Roumy,A.; Guillemot, C.; Alberi-Morel, M. L. Low-Complexity Single-Image Super-Resolution based on Nonnegative Neighbor Embedding. In Proceedings of the 23rd British Machine Vision Conference (BMVC), Guildford, UK, 3 September 2012; pp. 135.1–135.10.
  26. Zeyde,; Elad, M.; Protter, M. On Single Image Scale-Up Using Sparse-Representations. In Proceedings of International Conference on Curves and Surfaces, Avignon, France, 24-30 June 2010; pp. 711–730.
  27. Martin,; Fowlkes, C.; Tal, D.; Malik J. A Database of Human Segmented Natural Images and Its Application to Evaluating Segmentation Algorithms and Measuring Ecological Statistics. In Proceedings of Eighth IEEE International Conference on Computer Vision(ICCV), Vancouver, BC, Canada, 7-14 July 2001; pp. 416-423.
  28. Huang, B.; Singh, A.; Ahuja, N.Single Image Super-Resolution from Ttransformed Self-Exemplars. In Proceedings of IEEE Conference on Computer Vision and Pattern Recognition (CVPR), Boston, MA, USA,8-10 June 2015; pp. 5197-5206.
  29. Horé, ; Ziou, D. Image Quality Metrics: PSNR vs. SSIM.In Proceedings of International Conference on Pattern Recognition(ICPR), Istanbul, Turkey, 23-26 August 2010; pp. 2366-2369.
  30. Wang, Zhou.;Bovik,  C.; Sheikh, H. R.; Simoncelli, E. P. Image Quality Assessment: From Error Visibility to Structural Similarity. IEEE Transactions on Image Processing. 2004, 13, 600-612.
  31. He,; Zhang, X.; Ren, S.; Sun, J. Delving Deep into Rectifiers: Surpassing Human-Level Performance on ImageNet Classification. In Proceedings of IEEE International Conference on Computer Vision (ICCV), Santiago, Chile, 7-13 December  2015; pp. 1026-1034.
  32. He,; Zhang, X.; Ren, S.; Sun, J. Deep Residual Learning for Image Recognition. In Proceedings of IEEE Conference on Computer Vision and Pattern Recognition (CVPR), Las Vegas, NV, USA, 27-30 June 2016; pp. 1026-1034.
  33. Chowdhury, E. H.; Khandakar, A.;Mazhar, R.;Rahman, T.; Mahbub, Z. B. et al. Can AI Help in Screening Viral and COVID-19 Pneumonia? IEEE Access. 2020, 8, 132665-132676.

Point 2: Caption to Fig.7: the authors must add description for each panel (A)-(C) here

Response 2: Thanks for your suggestion, we add description for each panel (A)-(C). Previously our description was:

Figure 7. (a), (b) and (c) are the classification mixing matrix of CXR LR, SR and HR images respectively. (Vertical represents the real category, horizontal represents the prediction category, the right ratio matrix represents the accuracy rate of the four categories respectively, and the right ratio matrix represents the recall rate of the four categories respectively.)

This is not specific and sufficient, now we modify it as:

Figure 7. Classification mixing matrix of three kinds of images. (The vertical represents the true category, the horizontal represents the predicted category, the right ratio matrix represents the accuracy of each of the four categories, and the lower ratio matrix represents the recall rate of each of the four categories.) (A) is the classification result of CXR LR images. Since CXR LR images are obtained from CXR HR images by using the down-sampling method, there is a lack of information, resulting in the worst situation of correct classification. (B) Classification results of CXR SR images. The SMCPOS-SGD model is used to process the CXR LR image, and the details of the CXR LR image can be restored. The generated CXR SR image can effectively improve the number of correct classification of the four categories. (C) Classification results of CXR HR images. CXR HR images are the most primitive images, so it is best to classify them correctly.

Point 3:  Fig.4, caption: description for each panel (a)-(d) must be added here

Response 3: Thanks for your suggestion, we add description for each panel (a)-(d).Previously our description was:

Figure 4. PNSR, SSIM, train loss and eval loss of SGD and SMCPSO-SGD models were compared on Set5. (The black curve represents SGD optimization only, and the red curve represents SMCPSO and SGD combined training.)

This does not describe each panel specifically, now modify it as follows:

Figure 4. Comparison of SGD alone training and SMCPSO-SGD combined training in SET5. In (a, b), PNSR and SSIM of images generated during SGD and SMCPSO-SGD training are compared. The red curve is smoother than the black curve, and the PNSR and SSIM of the image are higher in each Epoch, indicating better quality of the generated images. In (c, d), the train loss and eval loss of red curve are always lower, and the black curve declines steadily in (c), but fluctuates greatly in (d), while the red curve declines steadily all the time, indicating that SMCPSO helps SGD better train.

Point 4:  Fig.3, caption: description for each panel (a)-(h) must be added here

Response 4: Thanks for your suggestion, we add description for each panel (a)-(h).Previously our description was:

Figure 3. Average error curve of multi-peak test functions F6, F8, F10, F12, F14, F18 and F20 with the number of iterations. (f(x) represents the optimal solution of the current iteration, and f(x*) represents the actual optimal solution of the function. The smaller f(x)-f(x*) is, the better the algorithm performance is.)

now modify it as follows:

Figure 3. Comparison of results for PSO, COBL and SMCPSO on multi-peak test functions. ( f(x) represents the optimal solution of the current iteration, and f(x*) represents the actual optimal solution of the function. f(x)-f(x*) is the current error represented by logarithm, and the smaller error is, the better the algorithm performance is.) (a) are the errors between the optimal value of each iteration and the actual optimal value of the f6 multi-peak test function by the standard PSO, COBL and SMCPSO methods respectively; Similarly,(b) is f8; (c) is f10; (d) is f12; (e) is f14; (f) is f16; (g) is f18; (h) is f20.

Reviewer 2 Report (Previous Reviewer 2)

The manuscript of Chaowei Zhou et al. describes the new approach of using the CNN for fast image reconstruction. I am glad that the authors have made the effort to improve this manuscript, especially since some of the previously presented results were incorrect. This contributed to improving the scientific quality of the manuscript. The organization and logical structure of the manuscript were improved. The references are actual and are related to the scope of the manuscript. The current presentation of the results confirms that the proposed approach offers some improvements in the classification of diseases based on X-ray images.

However, before acceptance of the manuscript for publication I suggest the extended  language, formatting, and editing errors corrections e.g.:

1) since 2 affiliations are indicated next to the authors' names (1,2), information on two institutions should be included in the manuscript, but there is still only a description of one institution and not two as it was indicated in the Author’s response.

2) additional language corrections are needed, e.g. in last review report I pointed out that,  “unclear sentences and statements e.g. line 180: “Table 2 indicates that the test results of SMCPSO-SGD on the general test  sets Set5[25], Set14[26] , BSD100 dataset[27] and Urban dateset[28]” – what indicates? Results? Table 2 presents/demonstrates the test results?

Author’s Response 5: I am very sorry that my mistake in wording caused you dyslexia. Thanks for your advice. I have modified the content as follows: Table 2 demonstrates that the test results of SMCPSO-SGD on the general test sets Set5 [25], Set14 [26] , BSD100 dataset [27] and Urban dateset [28].

But still, the word 'that' in this sentence is redundant if the authors only wanted to write that the results of the tests are presented in Table 2. The word 'that' implied that some description of the results is missing, e.g. Table 2 indicates/demonstrates  that the results of the study are similar, different, correlated, etc.

In abstract : ” In addition, chest X‐ray  super‐resolution images classification test experiment is conducted and experimental results  demonstrate the reconstruction effect of this model can improve the classification accuracy by 13.46%, among which the precision and recall rate of COVID‐19 are improved by 45.3% and 6.92%, 21 respectively”

It should be rather “In addition, chest X‐ray  super‐resolution images classification test experiment is conducted and experimental results  demonstrate the that reconstruction effect of this model can improve the classification accuracy by 20 13.46%, among which the precision and recall rate of COVID‐19 are improved by 45.3% and 6.92%, 21 respectively”

and much more…

3) lines 167-171: formatting errors in the equations.

Author Response

Response to Reviewer 2 Comments(The attachment can better express the formula, please see the attachment)

Point 1: since 2 affiliations are indicated next to the authors' names (1,2), information on two institutions should be included in the manuscript, but there is still only a description of one institution and not two as it was indicated in the Author’s response.

Response 1: Thank you for reminding us that we have added the information of a second institution in the latest manuscript.

Point 2.1: the word 'that' in this sentence is redundant if the authors only wanted to write that the results of the tests are presented in Table 2. The word 'that' implied that some description of the results is missing, e.g. Table 2 indicates/demonstrates  that the results of the study are similar, different, correlated, etc.

Response 2: Thanks for your instruction, I deleted "that" and amended it to:

Table 2 demonstrates the test results of SMCPSO-SGD on the general test sets Set5 [25], Set14 [26] ,  BSD100 dataset [27] and Urban dateset [28].

Point 2.2: In abstract : ” In addition, chest X‐ray  super‐resolution images classification test experiment is conducted and experimental results  demonstrate the reconstruction effect of this model can improve the classification accuracy by 13.46%, among which the precision and recall rate of COVID‐19 are improved by 45.3% and 6.92%, 21 respectively”

It should be rather “In addition, chest X‐ray  super‐resolution images classification test experiment is conducted and experimental results  demonstrate the that reconstruction effect of this model can improve the classification accuracy by 20 13.46%, among which the precision and recall rate of COVID‐19 are improved by 45.3% and 6.92%, 21 respectively”

Response 3: Thank you for pointing out another problem with my usage of "that". I add "that" in the designated position and modify it as follows:

In addition, chest X-ray super-resolution images classification test experiment is conducted and experimental results demonstrate that the reconstruction effect of this model can improve the classification accuracy by 13.46%, among which the precision and recall rate of COVID-19 are improved by 45.3% and 6.92%, respectively.

Point 3:  lines 167-171: formatting errors in the equations.

Response 3: Thanks to your suggestion, we have re-edited the equations in line 167-171.The revisions are as follows:

(Attachment visible)

Reviewer 3 Report (Previous Reviewer 3)

I wonder there is no change in this submission as compared to the previous one which was rejected from my side. So since, there is no change and my decision remains same as earlier.

Author Response

Response to Reviewer 3 Comments(For a better display of the formula, please see the attachment)

Point 1: The Introduction, Related Work and Methodology sections needs significant improvement. The methodology should be sufficient enough so that a reader may reproduce the results as well. 

Response 1: In response to your suggestions, we have made further improvements to the manuscript.

  • The following adjustments have been made in the introduction section :
  1. The research background of particle swarm optimization algorithm in convolutional neural networks is added. The details are as follows:

Tu et al [15] proposed an evolutionary convolutional neural network,  which uses ModPSO and backpropagation algorithm to train convolutional neural networks to avoid models falling into  local minima. This is the most advanced attempt, however,  the work does not optimize the particle swarm in terms of population diversity.

  1. The introduction of innovative points in the work of this manuscript is added in lines 51 to 62. The details are as follows:

In order to further explores the effect of PSO algorithm on the CNN training and SR image quality. In this paper, a CNN training method based on PSO algorithm is constructed, that is,  PSO algorithm is used to optimize CNN network parameters.   In addition, in view of the precocious problem of PSO,  the mutation induction of particles with high similarity is proposed according to the cosine similarity between  particles,  and the mutation probability decreases linearly with the number of iterations.   The cosine similarity mutation strategy  reinitializes the aggregated particles according to the cosine similarity,  which can maintain the better spatial solution distribution of the particle swarm.   Finally,  The model was used to perform SR on low-resolution chest X-ray (CXR) images and analyze the diagnostic effect of  pneumonia.   The CXR images classification experiment shows that although the hybrid model is trained on 91-image  dataset, it can also super-resolve CXR images effectively and enhance accuracy of classification.

  • The following adjustments have been made in the methods section:
  1. In order to repeat the Classification experiment of pneumonia, a section (3.3.Classification of pneumonia) was added in line 161-178 of the method section, in which the classification model and the pre-training model were explained. The details are as follows:

3.3. Classification of pneumonia

Deep convolutional neural networks are being used in medical diagnosis. ResNet34 [32] adopted the residual network structure to achieve a good balance between classification accuracy and network complexity. Therefore, ResNet34 [32] was selected as the diagnostic classifier for pneumonia. Since the pneumonia data set publicly available online is not large, transfer learning is used to train the model. ResNet34 [32] uses ImageNet weights and the full connection layer is modified to fit the four categories of the experimental data set.

Five indexes of accuracy, precision, sensitivity, F1 score and specificity [33] were used to evaluate the classification results of ResNet34 [32]. The calculation formula is as follows:

(Formula attachment visible)

where COVID-19, Lung Opacity, Normal and Viral Pneumonia for four class problem.

  1. In order to better introduce the method of using SMCPSO for mixed training. A detailed description of the flow diagram of the particle swarm optimization algorithm is supplemented in the manuscript in lines 155-166. The details are as follows:

3.2. FSRCNN model based on SMCPSO

In our implementation, we utilize the SMCPSO method to initialize the weights and bias of the FSRCNN model. The MSE is defined as the fitness function of SMCPSO, and the dimension of the particles is the number of parameters to be learned in FSRCNN network. Figure 2 illustrates the flowchart of the joint algorithm. The weight and bias of the FSRCNN model correspond to each dimension of the particle. The number of optimized particles is set to 50, and each particle represents a set of possible weights and biases of the FSRCNN model. The number of iterations is set to 10000. Every 100 iterations, the particle whose cosine similarity to the optimal particle is less than the average cosine similarity and whose random value is greater than the variation factor is initialized. Each iteration considers whether there is a better solution for the inverse particle of the particle, and if so, makes the particle its inverse particle. Until the iteration stop condition is reached, the particle swarm training part ends. Each dimension value of the optimal particle is corresponding to the weight and bias of the FSRCNN model, and the SGD algorithm is used to optimize the weight and bias of the model until the training is completed. The SGD algorithm is greatly affected by the initial position, so PSO can set the ideal initial position for SGD. Specifically, PSO is used to search the desired weights and bias of the CNN as the initial parameters of SGD algorithm. Higher accuracy can be achieved through this joint training method.

Point 2: I would also like to comment on the key contribution as the intuitiveness/novelty behind the propposed approach is quite insignificant. Authors should add key contributions highlighting the intuitiveness at the end of the introduction section. 

Response 2: The introduction of innovative points in the work of this manuscript is added in lines 51 to 62. The details are as follows:

In order to further explores the effect of PSO algorithm on the CNN training and SR image quality. In this paper, a CNN training method based on PSO algorithm is constructed, that is,  PSO algorithm is used to optimize CNN network parameters.   In addition, in view of the precocious problem of PSO,  the mutation induction of particles with high similarity is proposed according to the cosine similarity between  particles,  and the mutation probability decreases linearly with the number of iterations.   The cosine similarity mutation strategy  reinitializes the aggregated particles according to the cosine similarity,  which can maintain the better spatial solution distribution of the particle swarm.   Finally,  The model was used to perform SR on low-resolution chest X-ray (CXR) images and analyze the diagnostic effect of  pneumonia.   The CXR images classification experiment shows that although the hybrid model is trained on 91-image  dataset, it can also super-resolve CXR images effectively and enhance accuracy of classification.

Point 3: The related work section currently seems to be more like preliminaries. The overall work seems to be utilizaiton of existing building blocks to address the problem. 

Response 3: .

The overall work is indeed to optimize the training of the existing model FSRCNN by using particle swarm optimization algorithm, so as to achieve the purpose of optimizing the training results. Therefore, the convolutional neural network model, particle swarm optimization algorithm and some improvement strategies of particle swarm optimization in the field of image superresolution are mainly introduced in the relevant work part.

Round 2

Reviewer 1 Report (Previous Reviewer 1)

Now, the manuscript can be accepted

Reviewer 3 Report (Previous Reviewer 3)

Its improved as compared to earlier. Accepted from my side.

This manuscript is a resubmission of an earlier submission. The following is a list of the peer review reports and author responses from that submission.

Round 1

Reviewer 1 Report

The manuscript should be improved and revised, after which it can be reconsidered. Below are comments and suggestions that should help the authors improve it.

1) The whole reference list must be changed and reformatted. The authors must organize their reference list according to the MDPI style.

2) Space must be added before references (brackets [ ]) in the text. Reference numbers should be separated from words:   text [1,2]

3) Lines 49-58: it is recommended that the authors add 1-2 sentences which specify what original and novel their  manuscript presents. This will facilitate understanding by readers

4) Lines 28, 30:  the references must be presented as [2-4] and [1,7].

5) Figure 3: all the panels must be labeled with letters: (a), (b), (c), and so on. All then all the labels should be then described in the captions

6) Figure 4: all the panels must be labeled with letters: (a), (b), (c), and (d). All then all the labels should be then described in the captions

7) line 205: "evaluate system" ? What is meant here? Evaluated system? Evaluation system? Please clarify.

8) Figure 7: all the panels must be labeled with letters: (a), (b), and (c). All then all the labels should be then described in the captions

Reviewer 2 Report

The manuscript of Chaowei Zhou et al. is describing the new approach of using the CNN for fast image reconstruction. This is an interesting topic, but the preparation of the manuscript and the way the data are presented make it unsuitable for publication. The authors should prepare it more carefully before submitting it.

In general, the language is correct, but vague statements are quite frequent and editorial errors make it difficult to read all the way to the end of the manuscript.

The authors should also reconsider the logical structure of the manuscript.

The manuscript is written very briefly and in general terms, without discussing or explaining in detail the obtained results.

Editorial and formatting errors are common here and make this manuscript extremely difficult to read. Enormous number of formatting errors:

-double spaces (e.g.  line 9 in abstract and much more further),

- lack of spaces (see line 26 “resolution(LR)”, line 27: “imagesas”, line 28: “, etc[2]-[4]”, line 29 : “method[5]”  line 43: “et al.[18]used” line 72: “whereandare theLR” and much more further),

- unnecessary dots or in wrong places (e.g. line 61: “Dong et al[1].first” , line 68),

- sentences starting with a lower-case letter (e.g. line 63: “it preprocessed the images using bicubic…”, line 145: “for the sake of evaluating the search cap”)

Lines 18-21: “In addition, chest X-ray super-resolution images classification test experiment is conducted and experimental results demonstrate the reconstruction effect of this model can improve the classification accuracy by 13.46%, among which the precision and recall rate of COVID-19 are improved by 20 45.3% and 6.92%, respectively.”- this statement was not directly confirmed or indicated in the manuscript

The discussion of the results is residual and, in some cases, not present at all. Why include results that are not analyzed or discussed in any way in a paper? For example, there is no discussion of the results presented in Table 1,Table 3 or Fig.7. The authors present only 'empty' statements without explaining their basis, justification.

A lot of unclear sentences and statements e.g. line 180: “Table 2 indicates that the test results of SMCPSO-SGD on the general test 180 sets Set5[25], Set14[26] , BSD100 dataset[27] and Urban dateset[28]” – what indicates? Results? Table 2 presents/demonstrates the test results?

Line 183: ‘kingming initialization method”?-probably Kaiming

Some explanation of the used notation of functions (F6-F20)?- please provide it.

Lien 169: “Consistent with previous works, 91-image dataset[24]” – what kind of images?

I wonder why the F1 Score is the worst for the supper-resolution images? Generally, the F1 score tells you the model’s balanced ability to capture both positive cases (recall) and be accurate with the cases it does capture (precision). It should be higher for images with the highest resolution ( in this case SR), from which more features can be extracted? However, also in this case there is no comments and discussions about this result.

In Section 4.3, lines 191-219, the authors describe the methodology of evaluation/assessment of proposed approach, but these issues should be described in the section Methods.

Figures ( all subfigures of one figure) and their captions should be placed on the same page

Lack of affiliation of the second author ‘Aimin Xiong 2” in the manuscript

The abbreviation PSO is not explained in the abstract.

Reviewer 3 Report

The manuscript is not well written at all. The Introduction, Related Work and Methodology sections needs significant improvement. The methodology should be sufficient enough so that a reader may reproduce the results as well. 

I would also like to comment on the key contribution as the intuitiveness/novelty behind the propposed approach is quite insignificant. Authors should add key contributions highlighting the intuitiveness at the end of the introduction section. 

The related work section currently seems to be more like preliminaries. The overall work seems to be utilizaiton of existing building blocks to address the problem.